



# Contribution of horizontal and vertical advection to the formation of small-scale vertical structures of ozone in the lower and middle stratosphere at Fairbanks, Alaska

Miho Yamamori[1], Yasuhiro Murayama[2], Kazuo Shibasaki[3], Isao Murata[4], Kaoru Sato[5]

[1]Department of Teacher Education, Tsuru University, Yamanashi, 402-8555, Japan
[2]National Institute of Information and Communications Technology, Tokyo, 184-8795, Japan
[3]Department of Elementary Education, Faculty of Human Development, Kokugakuin University, Yokohama, 225-0003, Japan
[4]Graduate School of Environmental Studies, Tohoku University, Sendai, 980-8579, Japan
[5]Department of Earth and Planetary Science, University of Tokyo, Tokyo, 113-0033, Japan

*Correspondence to*: Miho Yamamori (yamamori@tsuru.ac.jp)

**Abstract.** The contribution of vertical and horizontal advection to the production of small-scale vertical ozone structures was investigated using data from an ozonesonde observation performed at intervals of 3 h in Fairbanks (64.8N, 147.9W), Alaska.

The dominant vertical scales of the ozone mixing ratio were determined to be 2–5 km, which were similar to those of horizontal winds and the temperature of the lower and middle stratosphere, using spectral analysis. Ozone fluctuations due to vertical advection were estimated from the potential temperature fluctuation and vertical gradient of the background ozone mixing ratio. Residual ozone fluctuations are attributed to horizontal advection. Fluctuations due to horizontal advection are dominant, as reported in previous studies. The cross-correlation of the effects of vertical and horizontal advection was also

evaluated. The correlation is relatively larger at altitudes of 18–23 km and 32–33 km compared to those at other height regions. In contrast to previous studies, horizontal advection by gravity waves seems to play a dominant role in the production of small-scale ozone structures at altitudes of 32–35 km.

## 1 Introduction

Vertical profiles of ozone in the lower stratosphere often display a small-scale layered structure (Dobson, 1973; Reid and
Vaughan, 1991). Because the photochemical lifetime of ozone in the lower stratosphere is at least several weeks, layered structures are considered to be produced by the advection of air from different latitudes or heights in which the ozone abundance differs.

Possible formation processes that may contribute to the layered structure are the characteristic large-scale flow structure and gravity waves. Gravity waves cause material surfaces to undulate with the same wavelength and period as those of
meteorological parameters (Danielsen et al., 1991; Teitelbaum et al., 1994, 1996). However, coexisting gravity waves do not have a sufficient amplitude to produce the observed amplitude of ozone fluctuation (Reid et al., 1994; Gibson-Wilde et al.,





1997; Tomikawa et al., 2002). The mechanism of large-scale flow producing the layered structure is considered to be the following: streamers associated with Rossby wave breaking are thinned through the action of vertical shear on small horizontal scales. In this case, the transport is quasi-horizontal along isentropic surfaces.

Ozone fluctuations were investigated in many previous studies in correlation with potential temperature fluctuations because both the ozone mixing ratio and potential temperature can be treated as tracers and increase monotonously with increasing height in the lower stratosphere (e.g., Teitelbaum et al., 1994). In such studies, ozone fluctuations that are in the same phase as potential temperature fluctuations are regarded to be induced by vertical displacements of material surfaces due to gravity waves. Pierce and Grant (1998) distinguished ozone fluctuations generated by Rossby waves from those produced by gravity

waves based on the magnitude of the cross-correlation coefficient between ozone and potential temperature profiles. Noguchi et al. (2006) estimated the ozone fluctuation due to vertical advection in a similar way to Teitelbaum et al. (1994). They estimated fluctuations due to horizontal advection by subtracting the ozone fluctuation due to vertical advection from the observed ozone fluctuation and separately investigated global statistics of the characteristics of ozone fluctuation due to vertical and horizontal advection. They reported that the contribution of horizontal advection is more dominant than that of

vertical advection, especially in the extratropics.

    On the other hand, horizontal advection by gravity waves has been considered to be less important for the production of a layered ozone structure compared to vertical advection. Gibson-Wilde et al. (1997) discussed the mechanisms of two laminated ozone structures observed at altitudes of 14 km and 18 km by using high temporal resolution data on both ozone and horizontal winds from lidar observations. They concluded that the laminated structure at 14 km was produced by large-

scale horizontal advection. Meanwhile, they mentioned the possibility that horizontal advection induced by gravity waves may contribute to the production of a laminated structure observed at 18 km. Recently, Ohyama et al. (2018) separately evaluated fluctuation components of ozone mixing ratios resulting from vertical displacements due to gravity waves, horizontal displacements due to gravity waves, and horizontal displacements due to Rossby waves in the height range below 30 km by ozonesonde observations at the southern tip of the South America. They showed that the ozone variability

associated with horizontal displacement due to gravity waves was an order of magnitude smaller than that caused by vertical displacement.

    In the present paper, a case is reported in which horizontal advection due to a gravity wave mainly contributes to the formation of a layered structure. The contribution of the horizontal and vertical advection to a layered ozone structure was examined utilizing high-resolution, high-frequency, and simultaneous observations of horizontal winds and the ozone

abundance.

    Because electrochemical concentration cell (ECC) ozonesonde observations require a couple of hours of preparation before the release of each balloon and for the balloon to burst, few high-frequency observation attempts were made in the past. In the present experiment, we performed intensive ozonesonde observation and obtained data with high temporal and vertical resolution over a 36 h period.


The description of the data used in the present paper is given in Section 2. Section 3 consists of the spectral analysis, evaluation of each contribution of vertical and horizontal advection, and inertia gravity wave analysis for a distinctive case. The results are discussed in Section 4. The summary and concluding remarks are provided in Section 5.

## 2 Data

The ozonesonde observation was conducted in Fairbanks (64.8N, 147.9W), Alaska, during August 23–30, 2003, as part of

the validation experiment of the "Alaska Project" carried out by the National Institute of Information and Communications Technology, Japan (Murayama et al., 2002), and the Improved Limb Atmospheric Spectrometer-II (ILAS-II) (Nakajima et al., 2006) onboard the Advanced Earth Observing Satellite-II (Yamamori et al., 2006). During the observation period, a total of 22 ECC ozonesonde observations were made including an observation synchronized with the ILAS-II near Fairbanks (conducted daily at 04 UTC) and intensified observations performed every 3 h (from 16 UTC on August 26 to 04 UTC on

August 28).

The En-Sci 2Z-GPS ozonesonde used in the experiment was equipped with a GPS receiver and provided horizontal wind velocity data based on positioning data, in addition to ozone concentration, temperature, and humidity data from the ground up to an altitude of 30 km. The ozone, temperature, and humidity data were transmitted during the flight at a 1.5 s interval. The time interval corresponds to the vertical interval of 7–8 m. The GPS receiver built into the polystyrene case of the ozone

sensor provided location data every 5 s. The data were linearly interpolated every 50 m. Furthermore, GPS radiosonde (Vaisala RS80-15GH) observations were carried out at 07, 10, and 13 UTC on August 26; 19 UTC on August 28; and 01 UTC on August 29. Although an observation at 01 UTC on August 27 failed to obtain horizontal wind data, we successfully observed the temperature, humidity, horizontal winds, and ozone from the surface to an altitude of 32 km. Table 1 shows the data obtained during each observation and used in the subsequent analyses.

Figure 1 presents the successive vertical profiles of the zonal and meridional wind, ozone mixing ratio, and temperature obtained from 04 UTC on August 26 to 04 UTC on August 29 used in the analyses. Each profile was shifted based on the time that had elapsed since 04 UTC on August 26.

During the observation period, the tropopause was located at a height of 9–10 km. The ozone mixing ratio is almost constant in the troposphere. It gradually increases with increasing height in the height region of 10–16 km and rapidly increases with

increasing height above 16 km. The largest vertical gradient was detected at altitudes of 20–25 km. Fluctuations with vertical scales of 2–3 km were observed above the tropopause.

The small-scale structures are clearer in horizontal wind components. Clear downward phase propagation was observed in the lower stratosphere, indicating an upward-propagating wave. The prevailing wind at the tropopause level is northward because of the upper-level low pressure west of Fairbanks.



## 3 Analysis

### 3.1 Vertical wavenumber spectra

We analyzed the vertical wavenumber spectra of the horizontal winds, temperature, and ozone mixing ratio of lower-stratosphere profiles (above an altitude of 15 km). The results are shown in Figure 2. Significant ozone peaks were detected at wavelengths of 3.6 km and 2.2 km. The horizontal wind components display significant peaks at wavelengths of 3.1, 1.8, and 1.4 km. Although no distinct temperature peaks were observed, a large spectral density was detected at a wavelength of ~2–5 km. We analyzed the filtered components extracted using the band-pass filter with cutoff wavelengths of 2 and 5 km. The range of the vertical wavelengths includes typical values of vertical wavelengths of gravity waves observed in the lower stratosphere (e.g., Murayama et al., 1992; Sato, 1994; Wang et al., 2005).

### 3.2 Ozone disturbances estimated from temperature disturbances

Assuming that the ozone mixing ratio is conserved, i.e., ozone fluctuations are due to the dynamical effect, the total (observed) ozone fluctuation $\chi'$ is the sum of ozone fluctuations due to vertical advection $\chi'_v$ and that due to horizontal advection $\chi'_h$:

$$\chi' = \chi'_v + \chi'_h \tag{1}$$

Consequently, the variance of ozone fluctuations can be written as follows:

$$\mathrm{Var}(\chi') = \mathrm{Var}(\chi'_v) + \mathrm{Var}(\chi'_h) + 2\mathrm{Cov}(\chi'_v, \chi'_h) \tag{2}$$

The last term indicates twice the covariance of $\chi'_v$ and $\chi'_h$.

The ozone fluctuations caused by vertical and horizontal advection were estimated using a method similar to that described in Noguchi et al. (2006). Because the potential temperature is conserved in adiabatic motion, the vertical displacement can be estimated as follows:

$$\delta z = -\theta'/\bar{\theta}_z \tag{3}$$

assuming a horizontally uniform background, where $\delta z$ is the vertical displacement and $\theta'$ and $\bar{\theta}$ are the perturbation and background ozone mixing ratio, respectively. The amplitude of the vertical displacement estimated with Eq. (3) is ten to a few tens of meters.

Then, $\chi'_v$ is estimated as follows:

$$\chi'_v = -\delta z \bar{\chi}_z = \theta' \bar{\chi}_z/\bar{\theta}_z \tag{4}$$

Next, $\chi'_h$ is estimated by subtracting $\chi'_v$ from the observed ozone fluctuation $\chi'$. Finally, $\chi'_v$ and $\chi'_h$ are substituted into Eq. (2).

The left panel of Figure 3 shows each contribution of the three terms of the right side of Eq. (2) to the total variance $\mathrm{Var}(\chi')$. The contribution of $\chi'_h$ is clearly dominant at any altitude, similar to findings in Noguchi et al. (2006). The contribution of $\chi'_v$ is largest (~0.2) at a height of ~20 km. There is the tendency that the $\mathrm{Var}(\chi'_v)$ and $2\mathrm{Cov}(\chi'_v, \chi'_h)$ cancel each other out.



The fact that the covariance of $\chi'_v$ and $\chi'_h$ is not zero implies that the two terms are not independent. The right panel of Figure 3 shows the vertical profile of the cross-correlation coefficients between $\chi'_v$ and $\chi'_h$. Relatively large values are distributed at heights of ~18–23 km heights and 32–33 km.

### 3.3 Time–height structure of the ozone fluctuation

Figure 4 shows the time–height sections of the vertically band-pass-filtered time-dependent (deviation from the 8 d mean) components of the zonal wind and ozone mixing ratio. Clear downward phase propagation seeming to be associated with inertia gravity waves was observed in the zonal and meridional (not shown) wind components throughout the upper troposphere and lower stratosphere, regardless of time. In the ozone component, on the other hand, layered structures frequently appeared in the lower stratosphere, persisting at almost the same altitudes. In addition, a clear phase descent with

time was observed at an altitude of 32–35 km from 16 UTC on August 26 to 04 UTC on August 28. The two sloping lines in the figures show the phase of the positive maximum of the westerly wind. The ozone components are almost in phase with the zonal wind component. Thus, the phase descent of ozone components was also suggested to be associated with the same inertia gravity wave as that of horizontal wind components.

### 3.4 Inertia-gravity waves fitting

We estimated the inertia gravity wave parameters for a distinctive case by fitting horizontal wind components to sinusoidal functions. The fitting procedure is the following: Firstly, the zonal wind component $u'$ and vertical wind component $v'$ are least-squares-fitted to plane wave solutions:

$$u' = u_{amp} \cos(mz - \omega t + \alpha) \tag{5}$$

$$v' = v_{amp} \cos(mz - \omega t + \gamma) \tag{6}$$

Subsequently, a combination of the vertical wavenumber $m$ and ground-based frequency $\omega$, amplitudes ($u_{amp}$, $v_{amp}$), and phases ($\alpha$, $\gamma$) giving the smallest residual sums of squares are selected. Next, the amplitudes and phases of vertical and meridional displacements are determined by utilizing dispersion and polarization relations of inertia gravity waves.

The case to which the fitting analysis was applied is the time–height cross section of $u'$ and $v'$ for the time duration from 19 UTC on 26 August to 13 UTC on 27 August and height region from 32 to 35 km. Zonal and meridional wind and ozone

mixing ratio fluctuations (Figs 5a–c) have monochromatic features. The fitted sinusoidal functions are displayed as parallel sloping lines in Figures 5e and f, superimposed on the shadings. The phase of the meridional displacement corresponding the parameters obtained above is shown in Figure 5g. Heavy solid and broken lines indicate the positive and negative maxima of the northward displacement. Because the ozone mixing ratio decreases with increasing latitude in the height region from 32 to 35 km, northward displacement leads to positive ozone fluctuation. The phase structure is in good agreement with the

ozone fluctuations, especially for the negative maxima observed at 34.2 km (04 UTC on August 27), 33.5 km (07 UTC on August 27), and 32.8 km (10 UTC on August 27).





On the other hand, the temperature component is characterized by the superposition of waves. The phase of vertical displacement associated with the inertia gravity wave is shown by parallel lines in Figure 5h. A heavy solid line indicating positive maximum of downward displacement corresponds to the decent of positive maxima of temperature at 34.5 km at 04

UTC 27 August and 33.8 km at 07 UTC 27 August. Other features are apart from plane wave solution estimated by the fitting analysis.

## 4 Discussion

The contribution of $\chi'_h$ is clearly dominant at any altitude, similar to findings in the previous studies. Most of the $\mathrm{Var}(\chi'_h)$ is considered due to large-scale flow (e.g. Ohyama et al., 2018). It is noteworthy that a relatively large cross-correlation

coefficient between $\chi'_h$ and $\chi'_v$ was determined at altitudes of 18–23 km and ~33 km. This suggests that the characteristics of atmospheric motion in such height regions that produce ozone fluctuations and the structure of background fields differ from that of other height regions with respect to ozone layer production.

The covariance of $\chi'_h$ and $\chi'_v$ at a height of ~33 km is relatively large and downward phase propagation of the ozone fluctuation can be clearly observed. A non-zero covariance implies that $\chi'_h$ and $\chi'_v$ depend on each other. This result does

not match the formation mechanism related to large-scale flow because the transport due to large-scale flow is quasi-horizontal along isentropic surfaces. On the other hand, in an inertia gravity wave, non-zero covariance is possible because a parcel oscillates along a slantwise surface.

Although gravity waves are dominantly observed throughout the lower stratosphere in the extratropics and tropics, ozone fluctuations due to gravity waves (evaluated from potential temperature fluctuations) are not dominant. This is because the

formation of layered ozone structures due to large-scale flow in the extratropics is more active than in the tropics (Noguchi et al., 2006). Another reason based on the present study is that dominant gravity waves are those with long periods, i.e., near-inertia oscillation, in the extratropics. The dominance of gravity waves with a near-inertial frequency was reported in several studies (e.g., Sato et al., 1999). The vertical displacement associated with such near-inertia gravity waves is relatively small. In the case shown in Figure 5, the aspect ratio of vertical to horizontal displacement is approximately 1:200. This implies

that the contribution of horizontal advection associated with such gravity waves is relatively large. Based on the ILAS-II version 1.4 data obtained near Fairbanks, the ratio of vertical to meridional components of the background ozone mixing ratio gradient at a height of 33 km was estimated to be ~100:1. Thus, the amplitude of ozone fluctuations due to horizontal advection associated with such gravity waves is possibly twice the amplitude of those due to vertical advection.




## 5 Summary and concluding remarks

The contribution of vertical and horizontal advection to the production of small-scale vertical structures of ozone abundance was investigated in this study by using a thorough ozonesonde observation conducted in Fairbanks, Alaska. The results can be summarized as follows:

  1. Spectral analysis revealed that fluctuations with 2-5 km vertical wavelengths dominate the ozone mixing ratio, horizontal winds, and temperature in the lower stratosphere.

2. Ozone fluctuation due to vertical advection was estimated based on the potential temperature fluctuation and vertical gradient of background ozone mixing ratio. The contribution of the vertical advection component to the total variation is the largest (~ 0.2) at a height of ~20 km.

  3. Ozone fluctuation due to horizontal advection was estimated by subtracting the vertical advection from the observed ozone mixing ratio. The contribution of horizontal advection is dominant throughout the lower stratosphere.

4. The cross-correlation of ozone fluctuations due to vertical and horizontal advection was evaluated. The correlation is relatively larger in height regions of 18–23 km and at ~33 km compared with other altitudes.

  5. A case is shown in which the horizontal advection by gravity waves mainly contributes to the production of the small-scale ozone structure.

  The fourth result suggests that the mechanisms producing the layered ozone structure and contributing to material transport

in the middle atmosphere differ depending on the altitude. It is necessary to perform more case studies and statistical analyses using data for large altitude ranges.

**Author contributions.**

  M. Yamamori, K. Shibasaki, and I. Murata designed the campaign observation and conducted the observations. Y. Murayama managed and coordinated the campaign observation. K. Sato contributed to the spectral and fitting analyses. MY

processed and analyzed the data and wrote the manuscript. All authors discussed the results and contributed to the final manuscript.

**Competing interests.**

  The authors declare that they have no conflict of interest.

**Acknowledgments.**

The ozonesonde campaign observation was performed as part of the "Alaska Project" carried out by the National Institute of Information and Communication Technology, Japan. The observation was carried out together with Seiji Kawamura,



Kensuke Yoshioka, Fumikazu Taketani, Tomoki Nakayama, Nao Ikeda, and Kazuyuki Miyazaki. We are indebted to William R. Simpson, Kenneth Sassen, and Richard L. Collins for their assistance with the observations at the University of Alaska, Fairbanks. All figures shown in this paper were produced by GFD-DENNOU Library.

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





Table 1: Data obtained during each observation and used for the analyses. No mark shows that the data were successfully obtained. The letter "N" means that the data were not observed. The letter "F" means failure to obtain the data.

| Obs. Time (yyyymmddhh) | Sensor | O3 | Hor. Wind | Temp. | Obs. Time (yyyymmddhh) | Sensor | O3 | Hor. Wind | Temp. |
|---|---|---|---|---|---|---|---|---|---|
| 2003082604 | ECC | | | | 2003082713 | ECC | | | |
| 2003082607 | GPS | N | | | 2003082716 | ECC | | | |
| 2003082610 | GPS | N | | | 2003082719 | ECC | | | |
| 2003082613 | GPS | N | | | 2003082722 | ECC | | | |
| 2003082616 | ECC | | | | 2003082801 | ECC | | | |
| 2003082619 | ECC | | | | 2003082804 | ECC | | | |
| 2003082622 | ECC | | | | 2003082816 | ECC | | | |
| 2003082701 | ECC | | F | | 2003082819 | GPS | N | | |
| 2003082704 | ECC | | | | 2003082822 | ECC | | | |
| 2003082707 | ECC | | | | 2003082901 | GPS | N | | |
| 2003082710 | ECC | | | | 2003082904 | ECC | | | |






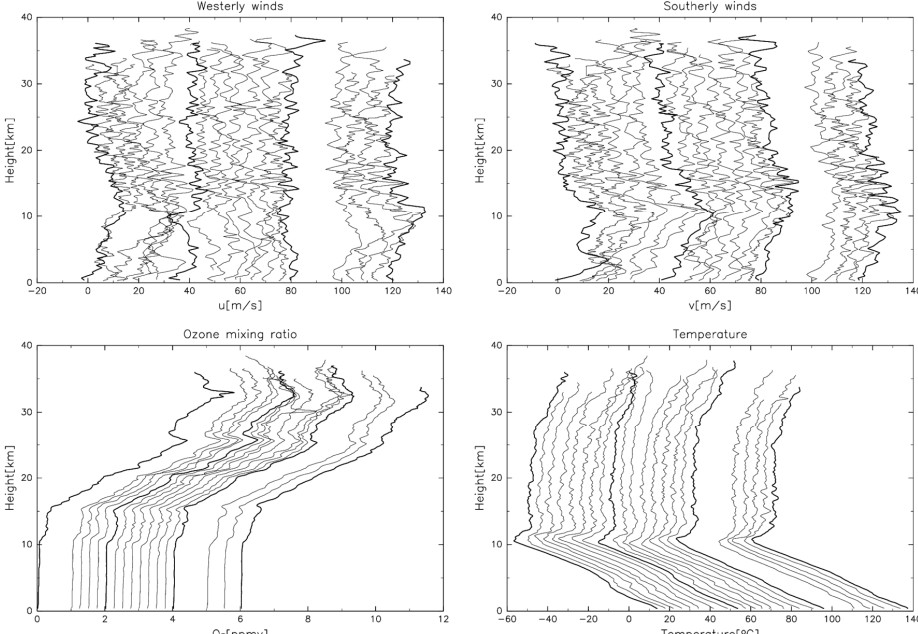

**Figure 1: Vertical profiles of the zonal and meridional wind, ozone mixing ratio, and temperature. Each profile was shifted based on the time that has elapsed since 04 UTC on August 26. The thick line represents the daily vertical profiles obtained at 04 UTC.**




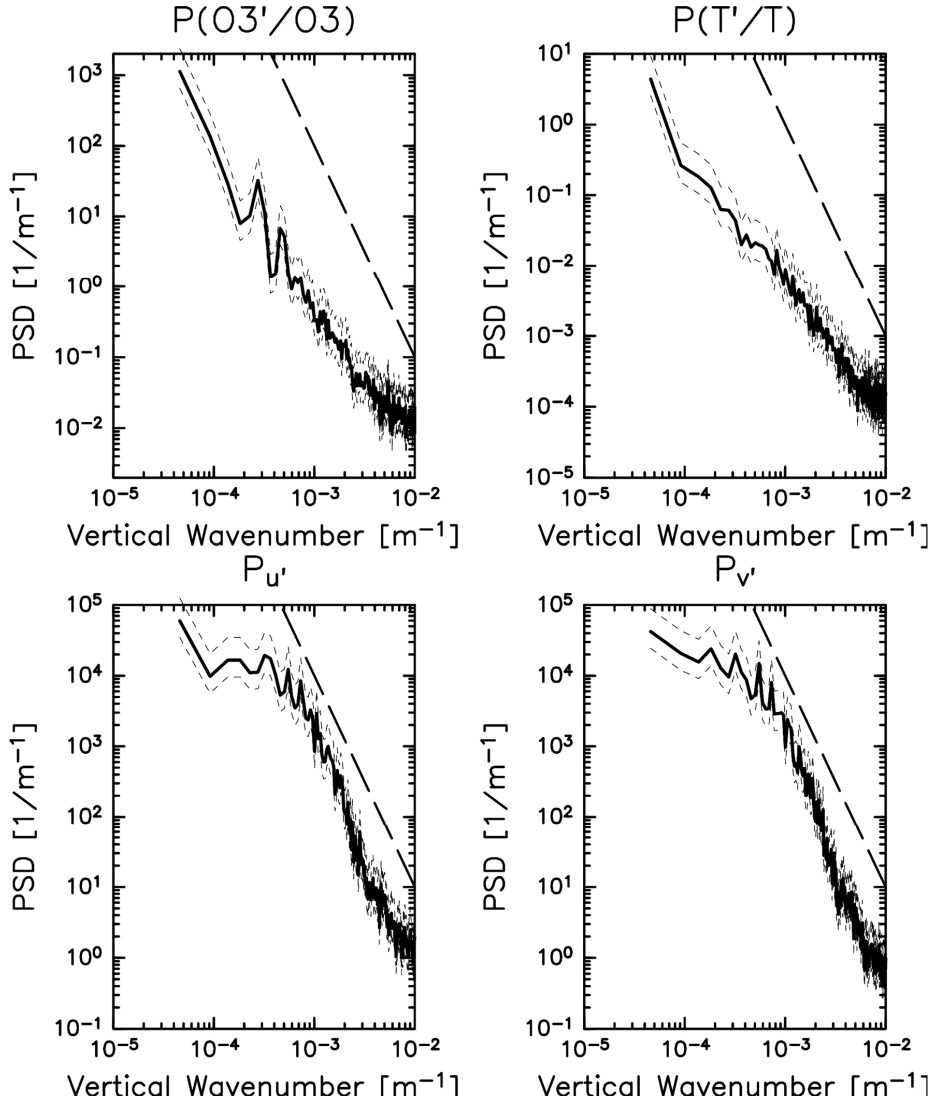

**Figure 2:** Vertical wavenumber ($m$) spectra for the relative ozone mixing ratio, relative temperature, and zonal and meridional winds. The two dotted lines show the 99% confidence limits. The dashed line indicates the slope of the spectra proportional to $m^{-3}$.






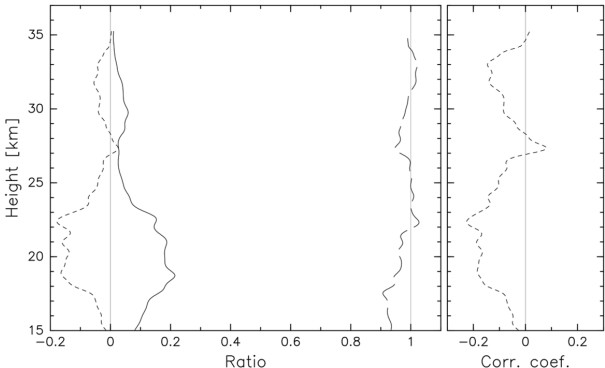

**Figure 3: (Left) Ratio of $\mathrm{Var}(\chi'_v)$, $\mathrm{Var}(\chi'_h)$, and $2\mathrm{Cov}(\chi'_h, \chi'_v)$ to the variance of the total ozone fluctuation $\mathrm{Var}(\chi')$ shown by solid, dashed, and dotted lines, respectively. (Right) Cross-correlation coefficients between $\chi'_h$ and $\chi'_v$.**





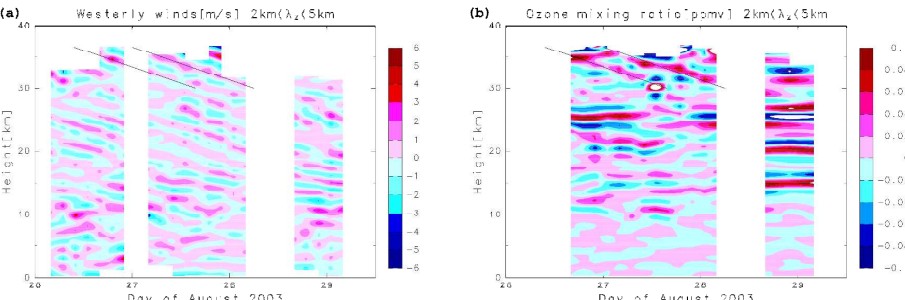

**Figure 4: Time–height sections of time-dependent components of the (a) zonal wind and (b) ozone mixing ratio extracted with a band-pass filter with cutoff wavelengths of 2 and 5 km from 04 UTC on August 26 to 04 UTC on August 29.**







**Figure 5: (Left) Time–height cross sections for the time duration from 19 UTC on August 26 to 13 UTC on August 27 and altitudes from 32 to 35 km. (a) Zonal and (b) meridional wind components, (c) ozone mixing ratio, and (d) temperature. (Right) Result of the fitting analysis. The slant lines show the phases of (e) zonal and (f) meridional wind components, (g) meridional displacement (heavy solid line indicates the maximum northward displacement), and (h) vertical displacement (heavy solid line indicates the maximum downward displacement) associated with the inertia gravity wave.**