# Peer review of "Contribution of horizontal and vertical advection to the formation of small-scale vertical structures of ozone in the lower and middle stratosphere at Fairbanks, Alaska"

_Atmospheric Chemistry and Physics, 2019_

## Referee Comment (RC1) · Anonymous Referee #1 · 6 Dec 2019

* General comments and major comments

This paper utilizes the high time-resolution (intervals of 3 h) observations of temperature, winds and ozone mixing ratio to reveal the contribution of vertical and horizontal advections to the production of small-scale vertical ozone structures in the lower stratosphere. Such a high temporal resolution observation would be feasible in principle but practically difficult. Therefore, the data obtained in this experiment are very precious not only to validate satellite measurements but also to probe the nature of the small-scale advections in the stratosphere.

[Figure]

However, additional analyses would be needed to make full use of the advantage of the high time resolution of the data. The authors should show new findings which had not been demonstrated in previous papers, e.g., Ohyama et al (2018). I suggest an idea for an additional analysis below.

In Figure 4(b), downward propagating phases appear even at the altitudes of 20-30km although their amplitudes are small. The authors may be able to distinguish ozone fluctuations due to gravity waves' horizontal advection and large-scale horizontal advection, since the ozone fluctuations due to large-scale horizontal advection should have much longer time-scale variation, and then remain at almost constant altitudes during the observation period. The separation of the two components would be a new analysis that has never done so far.

Here we assume that the observed ozone fluctuations X' are composed of three components, gravity waves' vertical advection, gravity waves' horizontal advection and large-scale horizontal advection:

X' = X'_GW_v + X'_GW_h + X'_Large_h.

The authors should distinguish X'_GW_h + X'_Large_h, as X'_GW_v was already estimated by eq. (4). Since the authors already obtained the components of horizontal wind fluctuations by a gravity wave, they can estimate the vertical distribution of the phase of ozone fluctuations due to a single gravity wave's horizontal advection for each profile. Although they cannot directly calculate the amplitudes of the ozone fluctuations because of the lack of the precise information on the background horizontal gradient of ozone mixing ratio, they would be able to estimate the amplitudes by fitting a sinusoidal curve as the downward propagating component (X'_GW_h) to the ozone fluctuations shown in Figure 4(b) after removing the constant or low degree-term (e.g., linear or quadratic) components (X'_Large_h) due to large-scale horizontal advection. Otherwise, they might be able to conduct a two dimensional (time and altitude) fitting in Figure 4(b) to extract X'_GW_h and X'_Large_h. In the analysis, potential temperature

would be appropriate for the altitude axis, since the large-scale advection occurs along isentropic surfaces.

Finally, I would also like to suggest to authors to ask English proofreading before submitting the final revised manuscript, as I could not understand some of the sentences very well.

* Specific comments

L30: "However, coexisting gravity waves do not have a sufficient amplitude to produce the observed amplitude of ozone fluctuation" I do not agree with this description, because the dominant mechanism of such layered structures generally depends on season and latitude. If needed, the authors should specify the seasons and latitudes focused on here.

L32: The authors could insert a sentence like "Large-scale flows can be large enough to cause the observed amplitude of ozone fluctuations." before the sentence "The mechanism . . .".

L35-45: Before this paragraph, the authors mentioned that the main mechanism of ozone layered structures was not gravity waves (because of their insufficient amplitudes) but large-scale flows. In this paragraph, however, the authors seem to emphasize the role of gravity waves (especially vertical advection) on ozone layered structures. Why did the authors focus on the role of gravity waves besides the role of large-scale flows here? Moreover, after this paragraph the authors showed the role of horizontal advection of gravity waves. Those descriptions would confuse readers. I suggest to the authors to show all the mechanisms that could produce ozone layered structures first of all at the beginning of the introduction section. Indeed, I found such a description in L52-53, which was almost at the end of the introduction section.

L59: Describe quantitatively how high the resolution and frequency of the data were. If those resolution and/or frequency were not achieved in the past studies, the authors

could also emphasize it here.

L61-64: The authors should not change the paragraph, because the descriptions in this paragraph are strongly related to the previous paragraph; the past studies would not have conducted such 3-h ozonesonde measurements because of the problem of ECC ozonesonde.

L61-64: The authors could also emphasize another advantage of the high-frequency observations: the separation of the contribution of gravity waves and large-scale flows in horizontal advection. As shown in Figure 4(b), the high-frequency observations clearly separated both components because of the different time scale of those components. The 3-h ozonesonde measurements resolved the downward propagations of the ozone fluctuations due to gravity waves above 30 km, while the fluctuations due to large-scale flows were almost constant. If the authors had not conducted high-frequency observations, they could not have found such phase propagations in ozone.

L88: The description on tropopauses can be included in the previous paragraph. The authors can also shortly describe the large-scale meteorological conditions together with the description of the tropopauses. Then, they can start a new paragraph from the sentence "The ozone mixing ratio. . .".

L97: The authors should describe how to obtain the vertical wavenumber spectra; otherwise they can refer to the paper(s) which had done a similar analysis. I think the authors calculated their spectra with a method like FFT applying some window function after removing large-scale trends from the original vertical profiles.

L109: How did the authors calculate the variance of ozone fluctuations? Did they calculate the variance at a given altitude level in all the profiles, or one by one profile?

L127: How did the authors calculate the cross-correlation coefficients? One by one profile?

L135: The values "32-35km" exceed the limitation described at L78.

L153-154: Refer to previous studies for the dependence of ozone mixing ratio on latitude and height. Otherwise, the authors should show a figure of the typical latitude-height cross section of ozone.

Section 4: It is slightly confusing that discussion is separated from the results described in Section 3. I do not think the section for discussion needs to be an independent section in this paper.

L164: I do not think that the cross-correlation coefficients of the values around 0.1-0.2 are significant.

L168-169: Why didn't any clear ozone fluctuation appear at altitudes of 18-23 km even though the cross-correlation coefficients at this altitude range were larger than around 33 km?

Figure 1: - Show the altitude of the tropopause in each vertical profile of temperature.

Figure 2: - Show the definition of relative ozone mixing ration and temperature. How did you calculate background values to derive relative values?

* Technical or minor comments

L25: "the photochemical lifetime of ozone in the lower stratosphere is at least several weeks" This needs references.

L28: "the layered structure" -> "the layered structures"

L28: "large-scale flow structure" -> "large-scale flows"

L30: "meteorological parameters" What exactly do you mean?

L33-34: "the action of vertical shear on small horizontal scales." I could not understand very well what authors meant. This sentence also needs references.

L51: I could not understand the meaning of "separately" here.

L78: I think the value "30 km" is not an exact value but an approximate one. The

authors should also show the reason why the data were limited around the altitude of 30km.

L81: ";" -> ","

L83: "an altitude of 32km" is not consistent with the value shown at L78.

L86-87: "Each profile was shifted based on the time that had elapsed since 04 UTC on August 26." The same sentence appears in the caption of Figure 1. Either of them should be deleted.

L89-90: "It gradually increases with increasing height in the height region of 10–16 km and rapidly increases with increasing height above 16 km." I could not understand very well what authors meant.

L92: "The small-scale structures are clearer. . ." than what?

L92: "Clear downward phase propagation was observed in the lower stratosphere" Show the range of the altitude and LT.

L94: "because of the upper-level low pressure west of Fairbanks." I would replace this by "because a low pressure system was located at the altitude of XX km in the west of Fairbanks".

L101: Put a phrase like "Based on the dominant wavelengths shown in the spectral analyses" before "We analyzed . . .".

L101: Replace "We analyzed the filtered components extracted using" by "We extracted the small-scale components using".

L102: I would move the sentence "The range of the vertical wavelengths . . ." to the end of "The results are shown in Figure 2." at L98.

L102: "The range" would be "The detected values", and "includes typical values" would be "were included in the typical range".

L112: "that" would be "the one".

L116: "delta z" should appear before "the vertical displacement" at L113 and the definition here should be deleted.

L117: "background ozone mixing ratio" would be "background components of potential temperature".

L117: The "typical" amplitude. . .

L117: ". . . is ten to a few tens of meters" would be "ranged from ten to a few tens of meters".

L153: Put "respectively" after "the northward displacement".

L165: I could not understand very well the sentence "This suggest. . .".

L183: I would say like "the same in the order of magnitude of the amplitude due to vertical advection" rather than "twice the amplitude of those due to vertical advection".
* * *

---

## Referee Comment (RC2) · Anonymous Referee #3 · 11 Feb 2020

**Summary:**

This manuscript discusses the contribution of horizontal and vertical transport processes induced by gravity waves to the formation of laminated structures identified in the lower and middle stratosphere from vertical profiles of ozone mixing ratio measured by ozonesondes over Fairbanks, Alaska. The authors estimate the ozone fluctuation caused by vertical advection from the vertical gradients present in the background profiles of potential temperature and ozone and the fluctuation of potential tempera-

ture. The difference between the observed and the estimated fluctuations of ozone is attributed to the horizontal advection. Based on the values of the variance and covariance between the estimated and the residual fluctuations of ozone, the author conclude that these fluctuations are dependent on each other and that the horizontal advection generating the residual fluctuation is dominant and is also caused by gravity waves.

**General comments:**

The topic addressed is relevant to the readers of Atmospheric Chemistry and Physics. There is a need for studies based on observations in order to advance our understanding of the dynamical processes regulating the distribution of ozone in the stratosphere. However, this manuscript suffers from several deficiencies. The authors have not made it clear what is original about the results presented. The relationships between the background gradients and the fluctuations of ozone and potential temperature induced by gravity waves have been documented for decades in several publications including some of the references cited in this manuscript. The authors argue that they quantified the contribution of vertical and horizontal advection to the vertical structure observed in the ozone profile. They claim that the horizontal advection is produced by gravity waves. However, the method and the analysis used the reach these conclusions are questionable as I describe in my main comments listed below. Given the substantial amount of work required to bring this manuscript up to an acceptable new piece of research, rejection is recommended.

**Specific comments:**

1. The ozone fluctuation due to the vertical advection is estimated from relation (4). This relation is very sensitive to the definition of the fluctuation of potential

temperature and the characterization of the background fields of ozone and potential temperature. The calculation of these important fields is not clear from the manuscript. In addition, the estimation of the contribution of the vertical advection to ozone in relation (4) uses the total fluctuation of potential temperature. Thus, it is implied that the temperature fluctuation is entirely due to the vertical motion. This assumption would be correct if the horizontal gradient of potential temperature on top of which gravity waves evolve is equal to zero. There is no argument in the manuscript supporting this assumption.

2. The authors argue that "The fact that the covariance of $\chi_v'$ and $\chi_h'$ is not zero implies that the two terms are not independent" (Line 126). They used this fact to rule out "the formation mechanism related to large-scale flow" (Line 170) and to conclude that the "non-zero covariance is possible because a parcel oscillates along a slantwise surface" (Line 171) in the presence of an inertia-gravity wave. Based on the method used to separate the horizontal and the vertical contributions, the dependence between $\chi_v'$ and $\chi_h'$ is not surprising. In fact, the authors impose relation (1) in deriving the contribution of the horizontal advection to ozone. However, this dependency does not necessary imply that $\chi_v'$ and $\chi_h'$ are physically dependent on each other through dynamics induced by gravity waves. The relative variance caused by the horizontal contribution alone seems to explain more than the total variance existing in the ozone profile! (22-25km and around 32km in Fig 3). I suggest that the authors analyze the vertical structure of the ozone profile using potential temperature as a vertical coordinate. Since ozone and potential temperature are both conserved under the influence of gravity waves, the wave-induced fluctuation in ozone should largely be removed in this coordinate. The remaining ozone fluctuations would indicate the contribution of transport induced by the large-scale flow along the main isentropes, with no perturbation in the potential temperature field.

3. The manuscript attributes at length the fluctuations of ozone and potential temperature to (inertia) gravity waves. However, little information is given on these waves. What are the vertical and horizontal wavelengths of these waves? What are their frequencies and how do they compare to the Coriolis parameter? What is the generation mechanism of these waves? How do you know that the fluctuations underlying the ozone fluctuations are indeed caused by gravity waves?

4. The spectra of ozone and temperature shown in Fig. 2 do not indicate common distinct peaks as would have been expected in the presence of gravity waves. The cross-sections presented in Fig. 4 indicate that the wave phases in the filtered ozone and the wind fields do not have the same tilts in most of the vertical range of the observations. Given that the group velocity of the waves propagates upward as implied by the downward phase propagation claimed above 30 km, why the ozone phases below 30 km do not show any sign of wave propagation? The authors claim that the two tilted lines in ozone "are almost in phase with the zonal wind component" (Line 136). This claim does not support the presence of gravity waves, as these waves would produce ozone fluctuations that are in quadrature of phase with respect to the wind field according to the linear theory of gravity waves. Also, the phase lines found in the filtered ozone above 30 km are questionable given that they are located near the upper bound of the data samples where the artifacts caused by the end effects of the filtering process are severe.

5. The authors state in Section 3.4 that "the amplitudes and phases of vertical and meridional displacements are determined by utilizing dispersion and polarization relations of inertia gravity waves" (Line 147). These relations are not explicitly given. More importantly, the wavelengths, the frequency and the background flow required for the dispersion relation are not clear. I understand that the authors derive the vertical wavelength from the observed vertical profiles. This wavelength, however, is not necessary the actual vertical wavelength of the waves. The balloon flies through the wave field during its ascent for more than 1.5 hours

(estimated from the values provided in lines 78-79). The wind field causes horizontal drifts of the balloon during its ascent. Given that the wave phases are generally tilted in space, the vertical wavelength apparent to the balloon is expected to be different from the actual vertical wavelength that would have been measured above a fixed location in the horizontal. Correction to the apparent vertical wavelength is not taken into account in the dispersion relation and the analysis presented in Section 3.4. This makes the conclusions drawn in this section incomplete and questionable.

---

## Author Comment (AC1) · 17 Apr 2020

We would like to thank the reviewers for their time in reviewing this work. Given the comments from the reviewers, we decided to withdraw this paper once. We would like to enhance our analysis based on suggestions from the reviewers and resubmit the contents.